# Low-Field Magnetic Stimulation Alleviates MPTP-Induced Alterations in Motor Function and Dopaminergic Neurons in Male Mice

**DOI:** 10.3390/ijms241210328

**Published:** 2023-06-19

**Authors:** Sathiya Sekar, Yanbo Zhang, Hajar Miranzadeh Mahabadi, Benson Buettner, Changiz Taghibiglou

**Affiliations:** 1Department of Anatomy, Physiology, Pharmacology, College of Medicine, University of Saskatchewan, 107 Wiggins Road, Saskatoon, SK S7N 5E5, Canada; sathiyasdr@gmail.com (S.S.); hmiranza@ualberta.ca (H.M.M.); bjb981@mail.usask.ca (B.B.); 2Department of Psychiatry, Royal University Hospital, 103 Hospital Drive, Saskatoon, SK S7N 0W8, Canada; yanbo.zhang@ualberta.ca

**Keywords:** MPTP, LFMS, neuron–glial functions, motor function, Parkinson’s disease

## Abstract

Recent studies show that repetitive transcranial magnetic stimulation (rTMS) improves cognitive and motor functions in patients with Parkinson’s Disease (PD). Gamma rhythm low-field magnetic stimulation (LFMS) is a new non-invasive rTMS technique that generates diffused and low-intensity magnetic stimulation to the deep cortical and subcortical areas. To investigate the potential therapeutic effects of LFMS in PD, we subjected an experimental mouse model to LFMS (as an early treatment). We examined the LFMS effect on motor functions as well as neuronal and glial activities in 1-methyl-4-phenyl-1,2,3,6-tetrahydropyridine (MPTP)-treated male C57BL/6J mice. Mice received MPTP injection (30 mg/kg, i.p., once daily for 5 days) followed by LFMS treatment, 20 min each day for 7 days. LFMS treatment improved motor functions compared with the sham-treated MPTP mice. Further, LFMS significantly improved tyrosine hydroxylase (TH) and decreased glial fibrillary acidic protein (GFAP) levels in substantia nigra pars compacta (SNpc) and non-significantly in striatal (ST) regions. LFMS treatment improved neuronal nuclei (NeuN) levels in SNpc. Our findings suggest that early LFMS treatment improves neuronal survival and, in turn, motor functions in MPTP-treated mice. Further investigation is required to clearly define the molecular mechanisms by which LFMS improves motor and cognitive function in PD patients.

## 1. Introduction

Parkinson’s disease (PD) is a progressive neurodegenerative disorder affecting approximately 1% of the population over the age of 60 worldwide [1]. PD is clinically characterized by slow or decreased movement, resting tremors, postural instability, and various other symptoms. It is primarily caused by the degeneration of dopaminergic neurons in the substantia nigra pars compacta (SNpc) and the depletion of dopamine in the striatum (ST) [2,3,4,5]. Unfortunately, currently approved treatments for PD decrease the symptoms but do not halt the disease progression. Most of the developed medications for rescuing dopaminergic neurons and restoring motor and non-motor functions have failed in clinical trials because of low efficacy or adverse effects [6]. As such, novel therapeutic strategies that prevent disease progression are urgently needed.

Repetitive transcranial magnetic stimulation (rTMS) is a non-invasive neurostimulation technique that is widely used as a tool for studying brain function and treating brain disorders such as major depression, PD, and Alzheimer’s disease [7,8,9]. rTMS generates brief and intense pulses of magnetic stimulation (peak magnetic field is around 1–2 Tesla, generating electric fields, E ≥ 100 V/m) to depolarize underlying neurons in focal brain areas [7]. A recent meta-analysis revealed that high frequency-rTMS (HF-rTMS) (≥5 Hz) treatment in the bilateral motor cortex (M1) improved both upper limb and walking performances in PD [8]. This finding was recommended as a treatment guideline for PD by a European expert consensus [10]. The same guideline also suggests that rTMS involving a larger cortex area and more sessions may further enhance treatment efficacy. In addition to motor function, the effect of rTMS in treating depression and cognitive deficits in PD has been studied [11]. The dorsolateral prefrontal cortex (DLPFC) is the most studied brain area for depression and cognitive deficit in PD. The evidence to support the efficacy of rTMS treatment for cognitive deficits remains incomplete [11,12]. Inconsistent results in support of rTMS are likely due to differences in treatment location as well as strength and duration of stimulation protocols [13]. In addition to rTMS, a few studies have explored the therapeutic potential of transcranial pulse electromagnetic field (T-PEMF) and transcranial direct current stimulation (tDCS) in PD [14,15]. Distinguished from conventional rTMS, T-PEMF and tDCS generate a low electromagnetic field below the action potential threshold [16]. Although these techniques have shown promising results, further studies are warranted to determine their true clinical impact on patients with PD. 

Low-field magnetic stimulation (LFMS), a new rTMS technology that produces low-intensity magnetic stimuli (10–50 Gauss (0.001–0.005 Tesla)), has shown rapid mood-elevating effects in patients with major depressive disorder (MDD) and bipolar disorder [17,18,19]. Importantly, LFMS produces diffuse magnetic stimulation with intermittent gamma bursts in cortical regions without the risk of inducing seizure or other side effects often seen in rTMS treatment, such as painful scalp sensations and headache [18,19,20,21,22,23]. In addition, LFMS treatment does not require hospitalization and patients can be treated anywhere. The intensity of magnetic stimulation with LFMS is also very low in comparison with rTMS (1–2 Tesla) technology and may, therefore, reduce the risk of side effects. In preclinical models of the disease, LFMS treatment has been shown to be neuroprotective [24,25]. Our previous work with LFMS in a mouse model of traumatic brain injury (TBI) also showed promising results, with LFMS treatment recovering motor and cognitive functions associated with the brain injury [26]. In cuprizone models of multiple sclerosis, LFMS has also been shown to promote myelin repair [27,28]. We, therefore, hypothesize that LFMS may improve motor functions and possess neuroprotective effects, supporting its use as a treatment strategy for patients with PD. In the present study, we examined the effect of LFMS on motor deficits and neuronal degeneration in a 1-methyl-4-phenyl-1,2,3,6-tetrahydropyridine (MPTP)-treated sub-chronic mouse model of PD. In addition, we studied the effect of LFMS on MPTP-induced astroglial and microglial activation in the SNpc and ST regions.

## 2. Results

### 2.1. LFMS Treatment Improved Motor Function in MPTP-Treated Mice

Motor function tests were performed for all the experimental mice. Beam walk and rotarod tests were studied 48 h after the last MPTP injection, and open-field locomotor and stride length were measured, on the next day. The values obtained from each test were collected and the differences between the groups were analyzed using one-way ANOVA followed by Tukey’s multiple comparison method as the post hoc test. Statistical results showing the impact of LFMS treatment (see Figure 1 for details) on the motor function of MPTP-treated mice are summarized in the next four sections.

#### 2.1.1. Beam Walk Test

A beam walk test was performed for all the experimental mice to assess motor coordination. The mice were pre-trained for the beam walk and then assessed 48 h after the last MPTP injection. MPTP mice took a longer time to traverse the narrow beam and their immobility period was increased, compared with the normal control mice (*p* < 0.001 and 0.01, respectively). Both the time taken to transverse the beam and the immobility period dropped significantly after LFMS treatment of MPTP mice (*p* < 0.001 and 0.01, respectively). The treatment of LFMS alone showed no significant change in the time taken and the immobility period compared with the normal control group (*p* = 0.989 and 0.999, respectively). The one-way ANOVA revealed a significant difference between the groups (time taken: F (3, 46) = 10.06; n = 12 for GI and GIV, and 13 for GII and GIII; *p* < 0.001 and immobility period: F (3, 46) = 6.489; n = 12 for GI and GIV, and 13 for GII and GIII; *p* < 0.001) (Figure 2A,B).

#### 2.1.2. Stride Length

In addition to the beam walk, a stride length test was also carried out for all the experimental mice to assess motor coordination. The average distance between the two successive forelimb and hindlimb footprints was measured. The average distance between the two successive limbs was significantly reduced in the MPTP mice compared with the normal control mice (*p* = 0.005 and 0.006, respectively), while the average distance was restored with the LFMS treatment (*p* < 0.001). The values obtained from mice treated with LFMS alone were found to be not significantly different from that of the normal control mice (*p* = 0.999). The one-way ANOVA results showed significant differences between the groups (forelimb: F (3, 46) = 9.241; n = 12 for GI and GIV and 13 for GII and GIII; *p* < 0.001; hindlimb: F (3, 46) = 8.707; n = 12 for GI and GIV and 13 for GII and GIII; *p* < 0.001) (Figure 2C,D).

#### 2.1.3. Rotarod Test

A rotarod test was performed 48 h after the last MPTP injection. Mice were allowed to walk on the rotating rod with an accelerated rotation rate of 4 rpm/s and the time taken by the mice on the rotating rod was recorded. MPTP mice were not able to walk on the rotating rod and fell within a short span of time (*p* < 0.001), while LFMS treatment in MPTP mice significantly improved (*p* < 0.001) the time taken to walk, indicating less motor deficit. Mice treated with LFMS alone performed similarly to that of the control mice (*p* = 0.720). The one-way ANOVA showed a significant difference between the groups (F (3, 46) = 23.680; n = 12 for GI and GIV and 13 for GII and GIII; *p* < 0.001) (Figure 2E).

#### 2.1.4. Open Field Locomotor Activity

Open field exploration showed that MPTP mice had reduced mobility during a 5 min observation period as evidenced by the decrease in both the total number of squares crossed, and the number of center squares crossed, in comparison with the normal control mice (*p* = 0.021, and *p* < 0.001, respectively), whereas the reduction in total distance travelled did not reach a significant level (*p* = 0.151). A significant increase in the immobility period was also observed (*p* < 0.001). MPTP mice treated with LFMS resulted in a non-significant (*p* = 0.064) increase in the total number of squares crossed (Figure 2F), and a significant (*p* = 0.011) increase in center squares crossed (Figure 2G). While the effect of LFMS treatment on the distance travelled by MPTP-treated mice (Figure 2H) was not significant (*p* = 0.898), it significantly decreased the immobility period (Figure 2I) compared with the MPTP mice (*p* < 0.001). The values obtained in mice treated with LFMS alone were found to be comparable to that of the control mice (*p* = 0.990, 0.863, 0.983, and 0.976, respectively). The one-way ANOVA analysis showed that except for the distance travelled (F (3, 46) = 1.972; n = 12 for GI and GIV and 13 for GII and GIII; *p* = 0.13), there were significant differences between the treatment groups (total number of squares crossed: F (3, 46) = 3.819; n =1 2 for GI and GIV and 13 for GII and GIII; *p* = 0.013, center squares crossed: F (3, 46) = 10.270; n = 12 for GI and GIV and 13 for GII and GIII; *p* < 0.001, and immobility period: F (3, 46) = 9.810; n = 12 for GI and GIV and 13 for GII and GIII; *p* < 0.001).

### 2.2. LFMS Treatment Significantly Improved TH and NeuN Levels in the SNpc and ST Regions of MPTP-Treated Mouse Brain

Immunohistochemistry of TH immune positive cells in SNpc (Figure 3A–D), the intensity of immunopositivity in ST (Figure 3F–I), and NeuN immunopositive cells in SNpc (Figure 4A–D) and ST (Figure 4E–H) regions were measured. The difference between the groups was analyzed using two-way ANOVA. A significant decrease in the percentage of TH and NeuN immunopositivity in SNpc (*p* < 0.001 for both TH and NeuN) and ST (*p* < 0.001 and *p* = 0.025, respectively) in regions of MPTP mouse brain were observed compared with the control mouse brain. LFMS treatment significantly improved these alterations in the SNpc (Figure 3E and Figure 4I) region by increasing TH and NeuN immunostaining, compared with the MPTP mouse brain (*p* = 0.007 and 0.003, respectively), while no significant difference was observed in the ST (Figure 3J and Figure 4I) region (*p* = 0.254 and 0.828, respectively). The mice treated with LFMS alone showed similar morphology to that of normal mice (*p* = 0.990). The two-way ANOVA between the groups showed significant differences for TH and NeuN (TH: F (3, 40) = 45.55; n = 6; *p* < 0.001, NeuN: F (3, 40) = 16.57; n = 6; *p* < 0.001), whereas the interactions between the SNpc and ST regions were not significant (TH: F (3, 40) = 0.680; n = 6; *p* = 0.570 and NeuN: F (3, 40) = 2.247; n = 6; *p* = 0.098).

The Western blotting results of the SNpc and ST regions between the groups are represented in Figure 3K and Figure 4J and the differences between the groups were analyzed using two-way ANOVA. MPTP mice showed a significant decrease in TH and NeuN levels (*p* = 0.002 and 0.003 for SNpc, 0.001 and 0.921 for ST regions, respectively) compared with the normal control mouse brain. A non-significant increase in TH (Figure 3L) and NeuN (Figure 4K) levels was observed in LFMS treatment when compared with the MPTP mouse brain (*p* = 0.087 and 0.222 for SNpc, 0.121 and 0.999 for ST regions, respectively). The values obtained in the LFMS alone treatment were comparable to that of normal control mice (*p* = 0.990). The two-way ANOVA between the groups showed a significant difference (TH: F (3, 40) = 19.66; n = 6; *p* < 0.001, NeuN: F (3, 40) = 6.114; n = 6; *p* = 0.002). The interaction between the SNpc and ST regions was found to be insignificant (TH: F (3, 40) = 0.283; n = 6; *p* = 0.837 and NeuN: F (3, 40) = 1.617; n = 6; *p* = 0.201).

### 2.3. LFMS Treatment Reduced GFAP Level, Thereby Suppressing Gliosis in MPTP-Treated Mouse Brain

GFAP immunopositive cells in the SNpc and ST regions are represented in Figure 5A–H, respectively. The differences between the groups were analyzed using two-way ANOVA. Measuring GFAP immunopositive cells showed a significant increase in GFAP positive cells in the SNpc and ST regions (*p* = 0.001 and *p* < 0.001, respectively) in MPTP mice when compared with the control mice. Compared with the MPTP mice, LFMS treatment significantly reduced these alterations in the SNpc region (*p* = 0.013), while a non-significant difference was observed in the ST region (*p* = 0.192) (Figure 5I). Mice brains treated with LFMS alone showed a morphology similar to the normal mice (*p* = 0.999). The two-way ANOVA analysis between the groups was found to be significant (F (3, 40) = 81.89; n = 6; *p* < 0.001, the interaction between SNpc and ST: F (3, 40) = 34.08; n = 6; *p* < 0.001).

The effects of LFMS on astroglial (GFAP) and microglial (IBA1) markers in the SNpc and ST regions of MPTP-treated mouse brains were measured using Western blotting (representative images in Figure 5J) and the differences between the groups were analyzed using two-way ANOVA. The MPTP mice showed significant increases in GFAP (*p* < 0.001 and *p* = 0.003, in SNpc and ST, respectively). While the IBA1 level was increased significantly in SNpc (*p* = 0.041) and the alteration in ST was insignificant (*p* = 0.448) when compared with the normal control mouse brain. The effect of LFMS treatment on GFAP in the SNpc and ST regions (Figure 5K; *p* = 0.998 and 0.408, respectively) and IBA1 (Figure 5L; *p* = 0.633 and 0.999, respectively) levels were insignificant when compared with the MPTP mouse brain. LFMS treatment of the control mice showed no effect when compared with the normal control mice (*p* = 0.990). The two-way ANOVA analysis between the groups was significant (GFAP: F (3, 40) = 25.29; n = 6; *p* < 0.001 and IBA1: F (3, 40) = 6.855; n = 6; *p* = 0.001). The interaction between the SNpc and ST regions was insignificant (GFAP: F (3, 40) = 2.822; n = 6; *p* = 0.051 and IBA1: F (3, 40) = 1.193; n = 6; *p* = 0.325).

### 2.4. Effect of LFMS Treatment on Caspase-3 Activation in SNpc and ST Regions of MPTP-Treated Mouse Brains

The data collected from the Western blot images of total and cleaved caspase 3 (represented in Figure 6A) were analyzed using two-way ANOVA (Figure 6B). A significant increase in cleaved-to-total caspase-3 ratio was observed in the SNpc and ST regions of the MPTP mice (*p* < 0.001 and *p* = 0.01, respectively) compared with those in the control mice. While LFMS treatment insignificantly decreased the caspase-3 ratio in SNpc (*p* = 0.055), it had no effect in the ST regions (*p* = 0.187). LFMS treatment of the control mice showed no effects on caspase 3 when compared with the untreated control mice (*p* = 0.999). The two-way ANOVA analysis was performed for the cleaved-to-total caspase-3 ratios and the F value between the groups was found to be F (3, 40) = 17.08; n = 6; *p* < 0.001. The interaction between the SNpc and ST regions was F (3, 40) = 0.376; n = 6; *p* = 0.771.

### 2.5. Effect of LFMS Treatment on Dopamine Level in the ST Region of MPTP Mouse Brain

The dopamine level in the ST region of the experimental mouse brain was analyzed using the ELISA technique and the differences between the groups were analyzed using one-way ANOVA. The dopamine level reduction in the MPTP mouse brains was not significant when compared with that of the control brains (*p* = 0.490). LFMS treatment appeared to increase the dopamine level in the ST region in comparison with the MPTP-treated mouse brain; however, it did not reach a significant level (*p* = 0.089). The mice brain treated with LFMS alone were comparable to that of the normal control mouse brain (*p* = 0.997). No significant differences between the groups were observed in the one-way ANOVA analysis (F (3, 16) = 2.189; n = 5; *p* = 0.129) (Figure 7).

## 3. Discussion

The present study provides the first evidence to demonstrate a healing effect of LFMS on an MPTP mouse PD model. LFMS treatment improved the motor function in MPTP mice as evidenced by improved performance in the beam walk, stride length, rotarod, and open field locomotor tests. Further, LFMS increased the TH level in the SNpc region, which may contribute to dopaminergic neuronal protection/recovery. LFMS treatment reduced astrogliosis (GFAP) in the MPTP-treated mouse brain. Based on the above observations, we suggest that LFMS plays a significant role in improving the dopaminergic neuronal functions, partly by regulating glial reactivity and motor functions in the MPTP-induced PD model; thus, it may be used as an early therapeutic choice in the management of PD.

Evidence supports that aberrant gamma oscillations linked to neurological diseases and movement-related changes in gamma amplitude were recorded in patients with neurological symptoms [29,30,31]. Researchers suggest that striatal gamma oscillation was decreased under low dopaminergic tone [32]. In the present study, the improving effect of motor function in MPTP mice treated with LFMS may be due to the modulation of gamma synchronization in neurons. Neuronal alterations in PD are not restricted to only dopaminergic neurons; however, the dopaminergic neurons in the nigrostriatal pathway degenerated to a greater extent than other neurons [33]. TH is the rate-limiting enzyme in dopamine homeostasis, and it has a direct link to PD pathogenesis [34,35,36]. Dong et al. [24] reported that low-frequency rTMS improved TH expression in an experimental mouse model of PD. In the present study, the observed increases in the levels of TH in SNpc and ST regions by Western blotting and TH immunostaining in LFMS treatment are corroborated by the above finding. LFMS treatment also improved the dopamine level in the ST region, which suggests a potential neuroprotective effect of LFMS on dopaminergic neurons in our MPTP mouse model of PD. Furthermore, the decreased cleaved-to-total caspase-3 ratio in LFMS treatment also suggests the protective effect of LFMS against apoptotic pathways.

The astrocyte–neuronal interlink plays a vital role in central nervous system homeostasis. The activation of astrocytes has been associated with various neurodegenerative diseases including PD [30]. Increased expression of GFAP, a biomarker representing astrocyte activation and astrogliosis, has been extensively reported in cellular and experimental animal models of PD [37,38]. In addition, microglia are activated during any neuronal insult. Neuroinflammation mediated by microglia is associated with the degeneration of dopaminergic neurons [39,40]. The activation of microglia releases reactive oxygen species and cytokines, leading to the death of dopaminergic neurons [41]. In the present study, the decreased levels of GFAP and IBA1 in the MPTP mice treated with LFMS suggest that LFMS improves neuronal function from astrogliosis and inflammatory insults. Immunohistochemical analysis showed that LFMS treatment decreased SNpc GFAP level compared with the MPTP-treated mouse brains, but the values were found to be non-significant in Western blotting. A similar result was observed in the ST region and increasing sample size may clearly define these differences. Reactive astrocytes have been reported to be involved in the neurological recovery of stroke models [42]. Astrogliosis under certain circumstances may lead to harmful effects. The observed changes in GFAP level may be beneficial or harmful to the neuronal circuits. However, LFMS decreased reactive astrocytes and inflammation which may be beneficial in the management of PD.

Repetitive transcranial magnetic stimulation (rTMS) has been extensively studied for the possible treatment of PD. Various clinical studies have suggested that rTMS plays a key modulatory role in improving motor functions in PD [43,44,45]. Matsumoto and Ugawa [46] suggested that rTMS exerted beneficial effects on motor symptoms by inducing synaptic plasticity in PD patients [47]. The beneficial effects of rTMS are also attributed to neurotransmitter restoration and improved synaptic plasticity [48]. A single TMS treatment improved the mRNA levels of neurotrophic factors and prevented neuronal death by enhancing neurogenesis. Moreover, studies showed that 8-week treatment with T-PEMF improved motor function and neuronal recovery through neurotrophic factors and anti-inflammatory effects in PD patients, but these treatments showed minimal effects and required other options [14,49,50]. Recently, we showed that LFMS, which is a non-invasive, simple, and laptop-like device, has promising results in the treatment of traumatic brain injury in mouse models [26]. The results obtained in the present study clearly show that LFMS treatment may be a beneficial treatment option for the management of PD. External magnetic field treatment may provide an alternative to invasive brain surgeries and may help PD patients in a more convenient way.

In conclusion, LFMS decreased GFAP levels and increased TH levels in MPTP mice, indicating that LFMS protects the dopaminergic neurons from MPTP insult. Additionally, LFMS improved motor function as evidenced by the beam walk, rotarod, stride length, and open-field locomotor tests in the experimental mouse model of PD, through unknown mechanisms. This study is the first evidence to show the therapeutic effect of LFMS and we anticipate that this non-invasive and portable method of treatment may benefit patients and provide new insight into the management of PD, without the side effects reported with TMS treatment. To take the results from bench to bedside, further detailed investigation on improving functions of dopaminergic neurons in both sexes is warranted to clearly elucidate the exact cellular events using different experimental models of PD. Although LFMS stimulation of the entire animal body showed promising results in the present study, the effect of LFMS on other parts of the body and their potential mitigating effects, if any, on the health of neurons inside the brain, will be studied in detail in our future investigations. Further studies are also warranted in elucidating the therapeutic effect of LFMS in PD treatment.

## 4. Materials and Methods

### 4.1. Chemicals and Reagents

MPTP was purchased from Sigma-Aldrich, Canada (Oakville, ON, Canada). Rabbit anti-tyrosine hydroxylase (TH; Cat# ab6211, RRID: AB_2240393) and mouse anti-neuronal nuclei (NeuN; Cat# MAB377, RRID: AB_2298772) were obtained from Abcam (Toronto, ON, Canada), and MilliporeSigma (Oakville, ON, Canada), respectively; rabbit anti-ionized calcium-binding adapter molecule 1 (IBA1; Cat# MA5-36257, RRID: AB_2890455) and mouse anti-glial fibrillary acidic protein (GFAP; Cat# MA5-12023, RRID: AB_10984338) were purchased from Invitrogen (Burlington, ON, Canada); rabbit anti-caspase-3 (Cat# 9662S, RRID: AB_331439), goat anti-mouse IgG (HRP linked; Cat# 7076S, RRID: AB_330924), and goat anti-rabbit IgG (HRP linked; Cat# 7074S, RRID: AB_2099233) were purchased from Cell Signaling (Beverly, MA, USA); goat biotinylated anti-mouse IgG (H&L; Cat# BA-9200, RRID: AB_2336171), goat biotinylated anti-rabbit IgG (H&L; Cat# BA-1000, RRID: AB_2313606), and the elite ABC-peroxidase kit (Cat# PK-6100, RRID: AB_2336819) were purchased from Vector Laboratories (Newark, CA, USA); mouse Anti-ß actin (C4) HRP (Cat# sc-47778 HRP, RRID: AB_2714189) was purchased from Santa Cruz Biotechnology (Dallas, TX, USA); and the dopamine (DA) ELISA kit (Cat# RD-DA-Ge) was obtained from Reddot Biotech Inc. (Kelowna, BC, Canada). All other chemicals and reagents used were of analytical grade.

### 4.2. Animals

Ten-week-old male C57BL/6J mice (Charles River, Montreal, QC, Canada; Cat # CRL: 027, RRID: IMSR_CRL:027) were used for the study. All the animals were housed in groups (2–5 animals/cage) under a 12 h light and 12 h dark cycle with temperature control (~21 °C) and were provided with standard food and water ad libitum. The experiment was approved by the University Animal Care Committee (UACC), University of Saskatchewan, Saskatoon, SK, Canada, and performed following the Guidelines of the Canadian Council on Animal Care (CCAC).

### 4.3. Experiment Design and Treatment

Animals were acclimatized to the laboratory conditions for 7 days before experimentation. Following acclimatization, animals were pre-trained for the beam walk test until the ceiling performance was reached. The animals were then randomly assigned to one of the following groups (12-13 mice/group): GI—control (n = 12), GII—MPTP (n = 13), GIII—MPTP + LFMS (n = 13), and GIV—LFMS (n = 12).

MPTP was administered to groups II and III, intraperitoneally (i.p.) at 30 mg/kg, once daily for 5 days. This sub-chronic dose regimen was selected as they induce the depletion of 90% of TH proteins and increase astrogliosis 24 h after injection [51]. Normal saline (NS, 0.9% NaCl) was administered i.p. once daily for 5 days to the control and LFMS groups. LFMS treatment was provided to the respective animals for a period of 20 min, 4 h after the first MPTP or NS injection, and then once daily for 6 days. Sham treatment was given to the control and MPTP groups by placing them on the machine for 20 min without any magnetic stimulation. The LFMS device (Beijing Antis Biotech Co., Ltd., Beijing, China) generates time-varying magnetic stimulation every 2 s, followed by an 8 s resting interval. The magnetic field changed every 2 min between the uniform and linear gradient. Each 2 s stimulation consists of 80 trains spiking 6 m/s at intervals of 19 m/s, which constitute the intermittent gamma burst stimulation at 40 Hz. The magnetic flux density was composed of six pulses with 0.13 m/s width and 1000 Hz frequency as described previously [51,52] (Figure 1).

At the end of the treatment period (48 h after the last MPTP injection), the mice were assessed for motor functions using beam walk and rotarod tests. Open-field locomotor activity and stride length were measured on the next day. Following the motor function tests (on day 8), 11 mice/group were sacrificed by cervical dislocation under deep anesthesia (5% isoflurane; the depth of anesthesia was confirmed by the observation of deep and shallow breathing and loss of withdrawal reflexes) and brains were isolated for dopamine measurement and Western blotting. The remaining mice were perfused, and brains were collected in 4% paraformaldehyde for immunohistochemistry [53].

### 4.4. Motor Function Analysis

#### 4.4.1. Beam Walk Test

A beam walk test was performed according to Sathiya et al. [51] with minor modification. Mice were pre-trained to traverse a narrow beam of 100 cm in length to reach an enclosed escape platform. A bright light (approximately 60 W) was placed above the narrow beam to create an aversive stimulus. This encourages the mice to traverse the beam to the dark enclosed goal box. Following the treatment, mice were placed individually at the start of the beam and analyzed for the time taken to run over the beam and the immobility period. The immobility period is the total time of all rests while traversing the beam. The maximum time given for a mouse to traverse the beam was 60 s and if the mouse did not reach the goal box in 60 s, the time taken was recorded as 60 s. The beam was cleaned with 70% alcohol between each animal. The observer who scored the test was masked to the treatment groups.

#### 4.4.2. Stride Length

Stride length was measured according to the method of Fernagut et al. [54]. The apparatus consisted of a runway, measuring 4.5 cm (w) × 40 cm (l) × 9.5 cm (h), and a dark goal box, measuring 20 cm (w) ×14.5 cm (l) × 6.5 cm (h), placed at one end of the runway. The box had a hole (45 mm in diameter) facing the runway. The runway was illuminated by a halogen lamp (approximately 60 W) so that mice placed on the runway would run toward the box. The fore and hind paws of the mice were wetted with non-toxic colored ink and placed at the other end of the runway, which was covered with a strip of white paper. The time taken by the mice to cross the runway was recorded and the stride length for both limbs was calculated as the distance between two fore paw prints (average of 3 consecutive readings) and the distance between two hind paw prints (average of 3 consecutive readings). The apparatus was cleaned with 70% alcohol between each animal. The observer who scored the behaviour was masked to the treatment groups.

#### 4.4.3. Rotarod

The rotarod test was performed according to Deacon’s method [55] with a slight modification. A five-chambered rotarod machine (Series 8; IITC Life Science, Woodland Hills, CA, USA) was used for the experiment. Following the treatment, mice were placed on the rod, with one in each chamber facing away from the direction of rotation. The rotation was started with an acceleration rate of 4 rpm/s. The time taken by the mice on the rotating rod was recorded. The test was repeated three times and the mean value for each animal was calculated. The maximum time and speed of the rotation were set as 120 s and 50 rpm, respectively. The apparatus was cleaned with 70% alcohol between each trial and the observer who scored the experiment was masked to the treatment groups.

#### 4.4.4. Open Field Test

Locomotor activity was evaluated by placing the mouse into one corner of the open-field arena (40 × 40 × 30 cm; the floor of the open field was divided into 16 equal squares) and observed for 5 min. The total distance travelled, immobility period(s), and number of total squares and center squares crossed were recorded. The floor of the maze was cleaned with 70% alcohol between each animal and the observer who scored the behaviour was masked to the treatment groups [56].

### 4.5. Western Blotting

The SNpc and ST regions [57] were isolated (n = 6) from the fresh brains and flash-frozen in liquid nitrogen. The tissues were then homogenized using lysis buffer (25 mM Tris, 150 mM NaCl, 0.1% sodium dodecyl sulfate, 0.5% sodium deoxycholate, and 1% Triton X-100, pH 7–8 with protease inhibitors: 1 mm PMSF, 10 μg/μL aprotinin, 10 μg/mL pepstatin A, 10 μg/mL leupeptin, 2 mm Na3VO4, 20 mm sodium pyrophosphate, 3 mm benzamidine hydrochloride, and 4 mm glycerol 2-phosphate). The supernatant containing total protein was determined by using the Bradford Assay with the DC Protein assay dye (#5000111, Bio-Rad, Mississauga, ON, Canada), boiled at 95 °C with 2X sample loading buffer (#1610737, Bio-Rad) for 5 min. The samples (40 µg of protein concentration) were then loaded and separated using sodium dodecyl sulfate polyacrylamide gel electrophoresis (SDS-PAGE) and transferred onto polyvinyl difluoride (PVDF) membranes. The membranes were blocked with 5% fat-free milk for 1 h at room temperature to block non-specific binding. The membranes were then washed with TBST (Tris-buffered saline, 0.1% Tween 20) 3 times, 5 min each. The target proteins were immunoblotted with primary antibodies (rabbit anti-TH (1:200; Cat# ab6211; RRID: AB_2240393; Abcam, Toronto, ON, Canada), mouse anti-NeuN (1:500; Cat# MAB377; RRID: AB_2298772; MilliporeSigma, Oakville, ON, Canada), mouse anti-GFAP (1:1000; Cat# MA5-12023; RRID: AB_10984338; Invitrogen), rabbit anti-IBA1 (1:1000; Cat# MA5-36257; RRID: AB_2890455; Invitrogen, Burlington, ON, Canada) and rabbit anti-Caspase 3 (1:1000; Cat# 9662S; RRID: AB_331439; Cell Signaling, Beverly, MA, USA) overnight at 4 °C and then with corresponding HRP-conjugated secondary antibodies (goat anti-mouse IgG (Cat# 7076S; RRID: AB_330924) and goat anti-rabbit IgG (Cat# 7074S; RRID: AB_2099233) HRP linked; Cell Signaling). Mouse Anti-ß actin (C4) HRP (1:5000; Cat# sc-47778 HRP; RRID: AB_2714189; Santa Cruz Biotechnology, Dallas, TX, USA) was used as the loading control and was incubated for 1 h at room temperature. The membranes were washed with TBST and exposed to enhanced chemiluminescence reagent (Bio-Rad) and then imaged using the ChemiDoc^TM^ MP Imaging System. Protein bands of interest were analyzed using NIH ImageJ software (https://imagej.net/, accessed on 2 June 2019) and expressed as the ratio of the target protein to β-actin [26].

### 4.6. Immunohistochemistry

Immunohistochemical evaluation of TH, NeuN, and GFAP expression in the SNpc and ST regions (n = 6 from randomly selected mice in the control and LFMS groups, and 7 from MPTP and MPTP + LFMS groups) were investigated using the Bachman method [58] with a slight modification. The perfused brains were collected and stored at 4 °C for 2–3 days in 4% paraformaldehyde. Sections measuring 30–40 µm thick through matched coronal levels of SNpc (sections at Bregma approximately −3.16 mm; interaural region 0.64 mm) and ST (sections at Bregma approximately 0.98 mm; interaural region 4.78 mm) were cut using vibratome and stored in cryoprotectant (25% glycerol, 30% ethylene glycol, 0.1% sodium azide, and PBS) at 4 °C until use. Four sections/regions/brains were selected in a random manner (the brains were sliced and placed one section per well (6-well plate), and the seventh section was placed into the 1st well; in this case, we used one section in every 6 sections). The sections were incubated with 0.3% hydrogen peroxide to remove endogenous peroxidase interference. The non-specific binding was blocked with 5% BSA for 1 h and then the sections were incubated with primary antibodies (rabbit anti-TH (1:100; Cat# ab6211; RRID: AB_2240393; Abcam), mouse anti-NeuN (1:100; Cat# MAB377; RRID: AB_2298772; Millipore), and mouse anti-GFAP (1:250; Cat# MA5-12023; RRID: AB_10984338; Invitrogen)) overnight. The next day, the sections were washed and incubated with secondary antibodies (goat biotinylated anti-mouse IgG (H&L; 1:500; Cat# BA-9200; RRID: AB_2336171) and goat biotinylated anti-rabbit IgG (H&L; 1:500; Cat# BA-1000; RRID: AB_2313606); Vector Labs) for a period of 1 h. Phosphate-buffered saline was used for washing between each step. The sections were stained with avidin–biotin substrate (VECTASTAIN^®^ Elite ABC-HRP Kit, Peroxidase; Cat# PK-6100; RRID: AB_2336819) and then with 0.05% DAB solution. All sections were mounted and counterstained with Mayer’s hematoxylin and visualized in a phase contrast trinocular compound microscope (OMAX Microscopes, Canada; Software: Toupview version 3.7 (http://www.touptek.com/product/showproduct.php?id=103&lang=en, (accessed on 1 June 2019), RRID: SCR_017998, ToupTek Photonics Co., Ltd., Hangzhou, China) at 10× magnification.

Quantification of TH, NeuN, and GFAP immunopositive cells in each section (6 animals per group, 4 non-consecutive sections per animal, and 1 section of every 6 was collected and used for each staining) were performed using NIH ImageJ software (https://imagej.net/, accessed on 2 June 2019, RRID:SCR_003070). The number of immunopositive cells per mm^2^ was counted at 100× magnification and the percentage of immunostaining of the target protein was calculated. The counting was performed randomly in 10 regions of the entire SNpc and ST regions and the mean value was calculated for each section. For TH in the ST region, the intensity of colour developed was measured and the percentage of immunostaining with that of the control was expressed. The person who performed immunostaining and scoring was masked to the treatment groups.

### 4.7. Enzyme-Linked Immunosorbent Assay (ELISA)

Experimental mice (n = 5) were euthanized via cervical dislocation under deep anesthesia and the ST region was isolated from each brain. The ST regions were processed for dopamine measurement using the ELISA kit (Reddot Biotech Inc., Kelowna, BC, Canada) following the manufacturer’s instructions. The homogenate for analysis was prepared using lysis buffer (25 mM Tris, 150 mM NaCl, 0.1% sodium dodecyl sulfate, 0.5% sodium deoxycholate, 1% Triton X-100, pH 7–8, and protease inhibitors). The intensity of colour developed was measured at 450 nm and is inversely proportional to the dopamine concentration. The minimum detectable level of dopamine using this kit was found to be less than 7.6 pg/mL. Various studies have measured dopamine levels using the HPLC technique. However, we measured dopamine levels in the ST region using ELISA as this method is found to be precise and accurate, and Nichkova et al. [59] validated ELISA for the measurement of dopamine content in urine samples.

### 4.8. Statistical Analysis

Data were expressed individually and were tested for the normality of distribution using the Shapiro–Wilk test. Differences between the groups were analyzed using GraphPad Prism (version 7.0, http://www.graphpad.com/, (accessed on 3 August 2019), RRID:SCR_002798). A *p* value of less than 0.05 was considered statistically significant. All data were tested for normality of distribution using the Shapiro–Wilk test. The differences between groups were analyzed using either one-way ANOVA (for the behavioural outcome and dopamine data) or two-way ANOVA (for TH, NeuN, GFAP, IBA1, and caspase-3 data) followed by Tukey’s multiple comparison method as the post hoc test with the SNpc and ST regions as factors. NIH ImageJ software was used to analyze the protein bands of interest in Western blots and the percentage of immunoreactivity in immunohistochemistry images. ANY-maze was used to measure the total distance travelled in the open-field test.

## Figures and Tables

**Figure 1 ijms-24-10328-f001:**
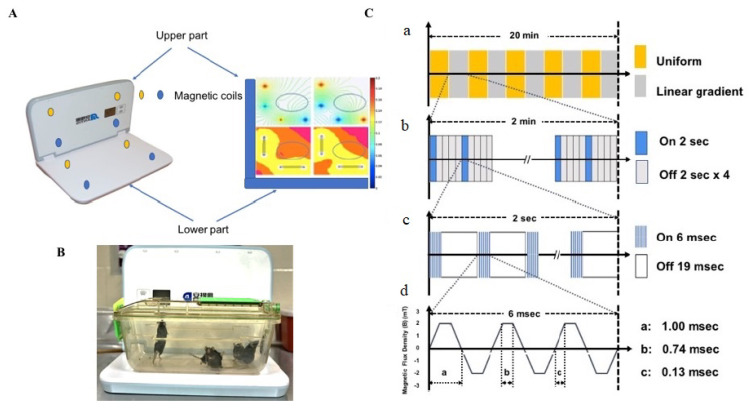
(**A**) Picture representing the LFMS device with magnetic field distribution; the magnetic coils circled with yellow and blue colour presented in all four corners horizontally and vertically generate the intermittent magnetic field; the current density is represented on the right side; (**B**) LFMS machine with animals in their home cage, which received non-invasive extremely low-field magnetic stimulation; (**C**) a schematic diagram representing the operational principle of the LFMS device. a. every 2 min the magnetic field is switched between uniform (magnetic flux density) and linear gradient distribution. b. Magnetic pulses: every 2 s of output is followed by an 8 s resting interval. c. Every 2 s of stimulation is composed of rhythmical trains spiking 6 ms pulses with intervals of 19 ms and constitutes the intermittent gamma burst stimulation at 40 Hz rhythm. d. Magnetic flux density.

**Figure 2 ijms-24-10328-f002:**
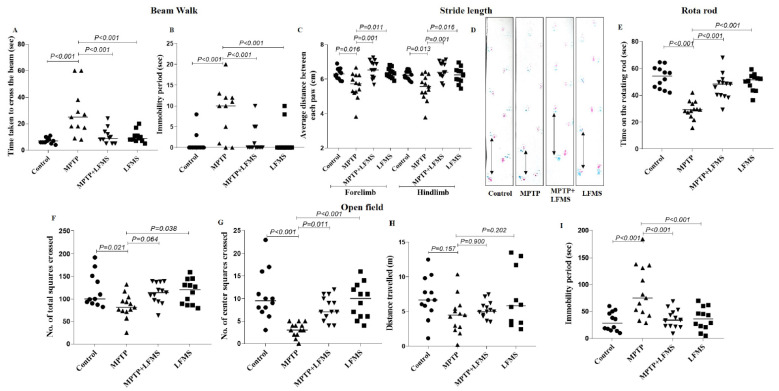
Effect of LFMS on functional outcome in MPTP mice. (**A**,**B**) The graphs represent the time taken to cross the beam(s) and immobility period(s), respectively, in the beam walk test; (**C**) the graph represents the average distance between the two successive fore or hind paws (average of three readings) in the gait test; (**D**) representative images of the foot paws of (1) control, (2) MPTP, (3) MPTP + LFMS, and (4) LFMS mice in the gait test. Colors show left (blue) and right (red) paw tracks; (**E**) the graph represents the time on the rotating rod(s) using the rotarod test; and (**F**–**I**) graphs representing the total number of squares crossed, number of center squares crossed, total distance travelled, and immobility period, respectively, in the open field test. n = 12 for the control and the LFMS groups and 13 for the MPTP and the MPTP + LFMS groups.

**Figure 3 ijms-24-10328-f003:**
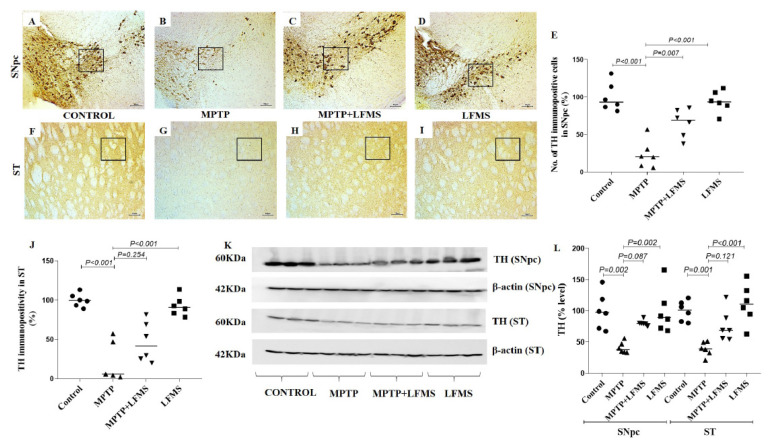
Effect of LFMS on tyrosine hydroxylase (TH) levels in the SNpc and ST regions of MPTP mice brain. Representative images of TH immunopositive cells in SNpc (**A**–**D**; at 4× magnification) and ST ((**F**–**I**); at 10× magnification) regions of control (**A**,**F**), MPTP (**B**,**G**), MPTP + LFMS (**C**,**H**), and LFMS alone (**D**,**I**) treated mice brain; boxes on the images indicate the regions where quantification was performed; (**E**) graph representing the percentage number of TH immunopositive cells in the SNpc region; and (**J**) TH immunopositivity in the ST region. (**K**) Representative Western blot images of TH levels and their respective loading control in the SNpc and ST regions of control, MPTP, MPTP + LFMS, and LFMS mice brain; and (**L**) graph representing the % levels of TH via Western blotting, n = 6.

**Figure 4 ijms-24-10328-f004:**
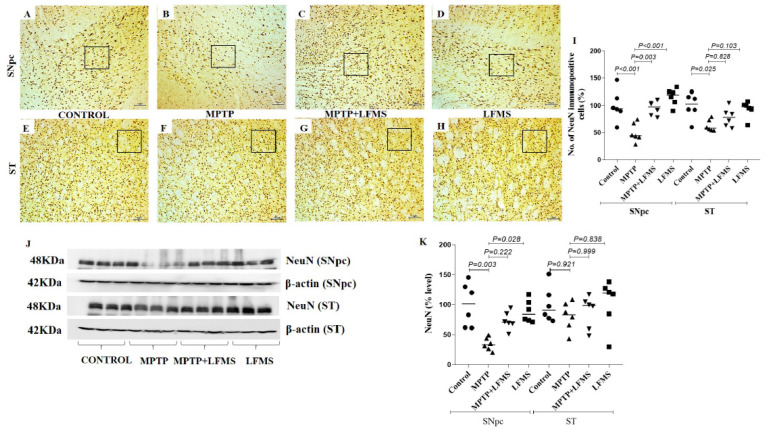
Effect of LFMS on neuronal nuclei (NeuN) levels in the SNpc and ST regions of MPTP-induced mice. Representative images of NeuN immunopositive cells in SNpc (**A**–**D**; at 4× magnification) and ST (**E**–**H**; at 10× magnification) regions of control (**A**,**E**), MPTP (**B**,**F**), MPTP + LFMS (**C**,**G**), and LFMS alone (**D**,**H**) treated mice brain; boxes on the images indicate the region where quantification was performed; and (**I**) graph representing the percentage number of NeuN immunopositive cells in the SNpc and ST regions. (**J**) Representative Western blot images of NeuN level and their respective loading control in the SNpc and ST regions of control, MPTP, MPTP + LFMS, and LFMS mice brain; and (**K**) graph representing the levels of NeuN via Western blotting, n = 6.

**Figure 5 ijms-24-10328-f005:**
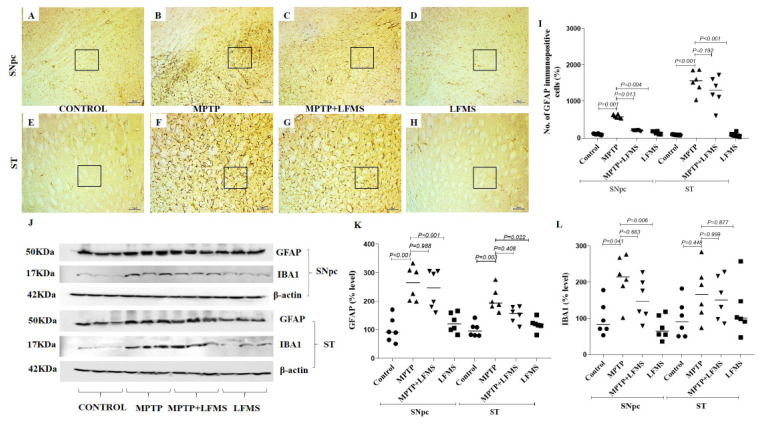
Effect of LFMS on glial fibrillary acidic protein (GFAP) level in the SNpc and ST regions of MPTP-induced mice. Representative images of GFAP immunopositive cells in SNpc (**A**–**D**) and ST (**E**–**H**) regions (at 40× and 100× magnification, respectively) of control (**A**,**E**), MPTP (**B**,**F**), MPTP + LFMS (**C**,**G**), and LFMS alone (**D**,**H**) treated mice brain; boxes on the images indicate the region where quantification was performed; and (**I**) graph representing the percentage number of GFAP immunopositive cells in the SNpc and ST regions. (**J**) Representative Western blot images of the GFAP and IBA1 levels and their respective loading control in the SNpc and ST regions of the control, MPTP, MPTP + LFMS, and LFMS mice brain. The graph represents the levels of GFAP (**K**) and IBA1 (**L**) via Western blotting, n = 6.

**Figure 6 ijms-24-10328-f006:**
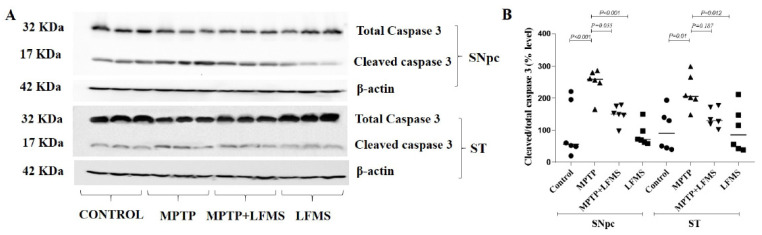
Effect of LFMS on caspase-3 level using Western blot analysis in the SNpc and ST regions of MPTP-treated mouse brains. (**A**) Representative Western blot images of cleaved and total caspase-3 levels and their respective loading control in the SNpc and ST regions of the control, MPTP, MPTP+LFMS, and LFMS mice brain, respectively; (**B**) graph represents the ratio of cleaved-to-total caspase-3; n = 6.

**Figure 7 ijms-24-10328-f007:**
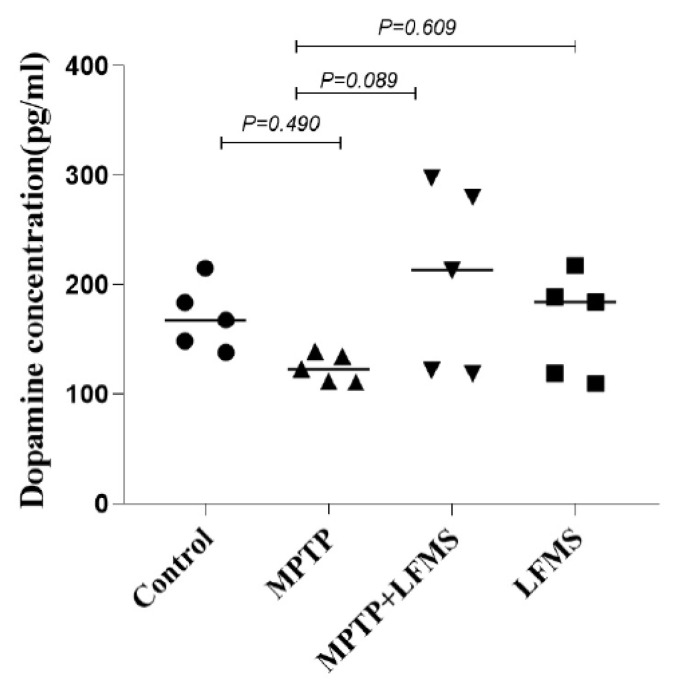
Effect of LFMS on striatal dopamine level using the ELISA technique in MPTP-treated mouse brains. The graph represents the concentration of dopamine in the ST region of the control, MPTP, MPTP + LFMS, and LFMS mice brain, respectively; n = 5.

## Data Availability

Data availability can be made upon request.

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
