# Peer review of "Low-Field Magnetic Stimulation Alleviates MPTP-Induced Alterations in Motor Function and Dopaminergic Neurons in Male Mice"

_ijms, 2023, doi:10.3390/ijms241210328_

Round 1

Reviewer 1 Report

Overall the paper is well written and clearly explained, and should be of interest to the scientific community.

minor edits: line 71 type; "injusry" spelling needs correction

Figures 2-5 have incorrect orientation; text is upside down.  Also, figure resoultion is too low, error bars and small text do not display legibly.

The effects of LFMS described are stronger for the histochemistry than for motor function effects.  As such, the authors should temper their conclusions in the discussion regarding the claims of restoring motor function.  

Author Response

Reviewer 1

Overall, the paper is well written and clearly explained, and should be of interest to the scientific community.

Comment 1: minor edits: line 71 type; "injusry" spelling needs correction.

Response: Sorry for the typing error. Corrected. Thank you.

Comment 2: Figures 2-5 have incorrect orientation; text is upside down.  Also, figure resolution is too low, error bars and small text do not display legibly.

Response: Thank you for the comment. We corrected the orientation of those figures in the submitted revised manuscript.

Comment 3: The effects of LFMS described are stronger for the histochemistry than for motor function effects.  As such, the authors should temper their conclusions in the discussion regarding the claims of restoring motor function. 

Response: To follow the respected reviewer’s comment, in the revised manuscript, we corrected the “restoring” to “improving” in the conclusion.

Reviewer 2 Report

Sekar et al. have looked at the effects of low-field magnetic stimulation in MPTP treated mice and found that this improved motor function and restored the number of dopaminergic neurons in the substantia nigra. The manuscript is of interest and provides more insight into the effect of magnetic stimulation and the role this might have in Parkinson's disease. I have several comments and suggestions, as per below. 

1. The authors conclude that their findings provide some evidence for the use of magnetic stimulation in people with Parkinson's disease. However, their findings are based on the reversal of symptoms and changes in the acute phase after MPTP treatment. How would these findings relate to the more chronic changes occurring in people with Parkinson's? 

2. How did the authors determine which mice to use for immunohistochemical evaluation? Was this randomly performed or did they use selection criteria (lines 476-477)?

3. Line 483: '... in a random manner'. Could the authors describe how this was performed?

4. Figure 3L: The mean for the control group does not seem to be at 100% which seems off given that these values represent % of levels in the control group. 

5. Line 71: 'injusry' should read 'injury'.

6. Lines 34-35: This sentence seems unrelated to any text before or after. What is the relevance of circadian disruption in this context or in the broader context of magnetic stimulation? 

Author Response

Reviewer 2

Sekar et al. have looked at the effects of low-field magnetic stimulation in MPTP treated mice and found that this improved motor function and restored the number of dopaminergic neurons in the substantia nigra. The manuscript is of interest and provides more insight into the effect of magnetic stimulation and the role this might have in Parkinson's disease. I have several comments and suggestions, as per below. 

Comment 1: The authors conclude that their findings provide some evidence for the use of magnetic stimulation in people with Parkinson's disease. However, their findings are based on the reversal of symptoms and changes in the acute phase after MPTP treatment. How would these findings relate to the more chronic changes occurring in people with Parkinson's? 

Response: The excellent point raised by the reviewer 2 is well taken. We have planned to extend the current study (acute PD animal model) to more chronic models of PD to investigate if we would observe any improvement in chronic sign/symptom changes.  

Comment 2: How did the authors determine which mice to use for immunohistochemical evaluation? Was this randomly performed or did they use selection criteria (lines 476-477)?

Response: The mice used for the immunohistochemistry evaluation was randomly selected for the different analysis. We believe that selecting randomly for the biochemical or histological analysis will provide us the precise results.

Comment 3: Line 483: '... in a random manner'. Could the authors describe how this was performed?

Response: We randomly selected the brain slices as mentioned below. We sliced the brain and collected one section in one well (6 well plate), the seventh section will be collected in 1st well. In this case, we used one section in every 6 sections. Also, the person who performed immunostaining and scoring was blinded to the treatment groups. The manuscript has been revised accordingly.

Comment 4: Figure 3L: The mean for the control group does not seem to be at 100% which seems off given that these values represent % of levels in the control group. 

Response: Yes, it’s correct. The values represented in figure 3L is the % levels in the control group. The figure and legends have been revised as per the comments.

Comment 5: Line 71: 'injusry' should read 'injury'.

Response: Sorry for the typing error. Corrected. Thank you.

Comment 6: Lines 34-35: This sentence seems unrelated to any text before or after. What is the relevance of circadian disruption in this context or in the broader context of magnetic stimulation? 

Response: Since sleep disturbance is one of the major issues with PD patients and directly linked to the circadian rhythm disruption, we mentioned it in the introduction. However as per the Reviewer’s comment, the sentence has been removed in the revised version.

Round 2

Reviewer 2 Report

I would like to thank the authors for their time and efforts in adjusting the manuscript in relation to the queries and concerns raised. I have no further comments.